# Genome-Wide Identification, Characterization and Expression Analysis of Plant Nuclear Factor (NF-Y) Gene Family Transcription Factors in *Saccharum* spp.

**DOI:** 10.3390/genes14061147

**Published:** 2023-05-25

**Authors:** Peter Swathik Clarancia, Murugan Naveenarani, Jayanarayanan Ashwin Narayan, Sakthivel Surya Krishna, Prathima Perumal Thirugnanasambandam, Ramanathan Valarmathi, Giriyapur Shivalingamurthy Suresha, Raju Gomathi, Raja Arun Kumar, Markandan Manickavasagam, Ramalingam Jegadeesan, Muthukrishnan Arun, Govindakurup Hemaprabha, Chinnaswamy Appunu

**Affiliations:** 1Division of Crop Improvement, Indian Council of Agricultural Research-Sugarcane Breeding Institute, Coimbatore 641007, India; swathikclarancia@gmail.com (P.S.C.); naveenamurugan03@gmail.com (M.N.); jashwinn89@gmail.com (J.A.N.); surikrish140@gmail.com (S.S.K.); prathima.pt@icar.gov.in (P.P.T.); r.valarmathi@icar.gov.in (R.V.); gs.suresha@icar.gov.in (G.S.S.); r.gomathi@icar.gov.in (R.G.); r.arun@icar.gov.in (R.A.K.); ghemaprabha1@gmail.com (G.H.); 2Bharathidasan University, Tiruchirappalli 620024, India; 3Department of Biotechnology, School of Life Sciences, Bharathidasan University, Tiruchirappalli 620024, India; manickbiotech@gmail.com; 4Centre for Plant Molecular Biology and Bioinformatics, Tamil Nadu Agricultural University, Coimbatore 641003, India; ramalingam.j@tnau.ac.in; 5Department of Biotechnology, Bharathiar University, Coimbatore 641046, India; arun@buc.edu.in

**Keywords:** *Saccharum*, *Erianthus arundinaceus*, plant nuclear factor, phylogenetic analysis, expression pattern

## Abstract

Plant nuclear factor (NF-Y) is a transcriptional activating factor composed of three subfamilies: NF-YA, NF-YB, and NF-YC. These transcriptional factors are reported to function as activators, suppressors, and regulators under different developmental and stress conditions in plants. However, there is a lack of systematic research on the *NF-Y* gene subfamily in sugarcane. In this study, 51 *NF-Y* genes (*ShNF-Y*), composed of 9 *NF-YA*, 18 *NF-YB*, and 24 *NF-YC* genes, were identified in sugarcane (*Saccharum* spp.). Chromosomal distribution analysis of *ShNF-Ys* in a *Saccharum* hybrid located the *NF-Y* genes on all 10 chromosomes. Multiple sequence alignment (MSA) of ShNF-Y proteins revealed conservation of core functional domains. Sixteen orthologous gene pairs were identified between sugarcane and sorghum. Phylogenetic analysis of NF-Y subunits of sugarcane, sorghum, and *Arabidopsis* showed that ShNF-YA subunits were equidistant while ShNF-YB and ShNF-YC subunits clustered distinctly, forming closely related and divergent groups. Expression profiling under drought treatment showed that NF-Y gene members were involved in drought tolerance in a *Saccharum* hybrid and its drought-tolerant wild relative, *Erianthus arundinaceus*. *ShNF-YA5* and *ShNF-YB2* genes had significantly higher expression in the root and leaf tissues of both plant species. Similarly, *ShNF-YC9* had elevated expression in the leaf and root of *E. arundinaceus* and in the leaf of a *Saccharum* hybrid. These results provide valuable genetic resources for further sugarcane crop improvement programs.

## 1. Introduction

Plant growth, development, and biomass production are severely affected by various biotic and abiotic stresses. Plants exert various stress-responsive mechanisms to combat these stresses. These defensive mechanisms are in turn regulated by various genes. Transcription factors are renowned for their regulatory role in modulating gene expression [1,2]. Regulation of gene expression by transcription factors assists in modulating downstream signaling pathways [3]. Plant nuclear transcription factor (NF-Y, Heme activated protein (HAP), or CCAAT binding factor (CBF)) is a transcription factor that binds to the CCAAT box sequence in the promoter region and modulates vital transcription machinery regulating various developmental and stress-responsive pathways [4,5].

NF-Y is a trimeric complex composed of the subunits NF-YA (HAP2/CBF-B), NF-YB (HAP3/CBF-A), and NF-YC (HAP5/CBF-C). NF-Ys regulate essential processes as individual subunits and as a complex. They also play a regulatory role by interacting with other transcriptional factors [6]. The NF-Y complex is formed by heterodimerization of subunits NF-YB and NF-YC in the cytoplasm, followed by heterotrimerization of NF-YA with the NF-YB–NF-YC dimer in the nucleus. This trimeric complex binds with high specificity to the CCAAT region of DNA to regulate transcription machinery [4,7,8,9] acting as activators [9] or repressors [7,8]. In the plant genome, multiple gene members encode for NF-Y subunits, resulting in an influx of *NF-Y* gene family members. The number of *NF-Y* genes and their copies varied in different plant species [10,11,12]. In addition, these transcription factors are reported to modulate gene functions at post-transcriptional levels [13]. *NF-Y* overexpression in plants resulted in enhanced developmental processes, such as embryogenesis [14], seed germination [15], flowering time [16], primary root elongation [17], photosynthesis [18], endosperm development [19], and photomorphogenesis [20], and improved tolerance to drought [21,22], salinity [23], and osmotic [24] stresses.

Sugarcane is an economically important crop that is the major source of raw material for the food and biofuel industries. Sugarcane yield is significantly affected by biotic and abiotic stresses, with drought stress being the most important constraint in sugarcane production [25,26]. Considerable attention has been given to developing drought-tolerant varieties to sustain sugarcane yield under drought conditions. It is important to understand abiotic stress-tolerant mechanisms, which involve intricate regulatory networks and pathways. NF-Y transcriptional factors are proven to play an essential role in tuning various stress-responsive mechanisms [5]. Hence, this study aims to identify and characterize members of the *NF-Y* gene family in the *Saccharum* hybrid genome. The current study also intends to understand the potential role of *NF-Y* genes under drought stress by studying their expression patterns in a *Saccharum* hybrid and their drought-tolerant wild relative, *Erianthus arundinaceus*.

## 2. Materials and Methods

### 2.1. Identification of NF-Y Gene Members in Sugarcane

The sugarcane genome was searched for *NF-Y* genes using conserved regions of *Sorghum bicolor NF-Y* gene sequences, which is a close relative of sugarcane. The obtained sequences were further used as queries to find *NF-Y*s in sugarcane “(https://sugarcane-genome.cirad.fr/ (accessed on 20 March 2022)”. Retrieved sequences were checked for core NF-Y domains using InterProScan “https://www.ebi.ac.uk/interpro/search/sequence/ (accessed on 1 April 2022)”, MotifSearch “https://www.genome.jp/tools/motif/ (accessed on 7 April 2022)”, and NCBI’s Conserved Domain Database (CDD) “https://www.ncbi.nlm.nih.gov/Structure/cdd/wrpsb.cgi (accessed on 14 April 2022)”. Redundant and incomplete sequences were removed. Identified NF-Y sequences (*ShNF-Y*) were located on the chromosomes using BLAST and manually marked. Gene numbers were assigned for *ShNF-Y* genes based on their physical positions on the chromosome. *ShNF-Y* sequences were predicted for open reading frames (ORFs), coding sequences (cds), and proteins. A bioinformatic pipeline for the identification, in silico and expression analyses of *NF-Y* genes is given in Figure 1.

### 2.2. Physiochemical Properties and Subcellular Localization Prediction

Physiochemical properties of ShNF-Y proteins (molecular weight, isoelectric point (pI), and GRAVY (grand average of hydropathicity)) were predicted using the Protparam tool “https://web.expasy.org/protparam/ (accessed on 1 July 2022)”. Subcellular localization of ShNF-Y proteins and gene ontology were predicted using the LocTree3 web server “https://rostlab.org/services/loctree3/ (accessed on 10 August 2022)”.

### 2.3. Multiple Sequence Alignment and Phylogenetic Analysis

Conserved core regions of ShNF-YA, ShNF-YB, and ShNF-YC proteins were aligned with conserved regions of sorghum and *Arabidopsis* NF-Y subunits using Unipro UGENE v46.0. Phylogenetic analysis was carried out using MEGA 10.0.5. Multiple sequence alignment was carried out using MUSCLE with parameters (Gap Open-2.90, Gap Extend 0.00, Hydrophobicity Multiplier 1.20, Max Memory in MB 2048, Max Iterations 16, Cluster Method UPGMA, and Min Diag Length (Lambda) 24). Phylogenetic analysis was carried out with 1000 bootstrap replicates and with the following parameters: Substitutions type—Amino acid, Model—Poisson model, Rates among sites—Uniform rates, Pattern among lineages—Same (homogeneous), Gaps/Missing data treatment—Pairwise deletion, Number of threads—7.

### 2.4. Chromosomal Distribution, Gene Structure, and Synteny Analysis of ShNF-Y Genes

Gene structures of ShNF-Y genes were predicted using Gene Structure Display Server 2.0 “http://gsds.cbi.pku.edu.cn/ (accessed on 15 May 2022)”. Gene and coding sequences of the respective ShNF-Y gene members were uploaded to predict gene structures. Collinearity analysis of 18 *NF-Y* sorghum genes was carried out using SynFind “https://genomevolution.org/coge/SynFind.pl (accessed on 20 June 2022)”. Synteny blocks of *NF-Y* genes were identified in *Arabidopsis* and rice. Ideograms were generated using Circos “http://circos.ca/ (accessed on 25 June 2022)”.

### 2.5. Identification of Conserved Motifs in ShNF-Y Genes and Proteins

Conserved motifs in ShNF-Y genes and proteins were determined using the MEME suite “http://meme-suite.org/tools/meme (accessed on 10 February 2023)”. The site distribution was set to any number of repetitions and number of motifs to be identified was set to 10.

### 2.6. Three-Dimensional Structure Prediction of ShNF-Y Proteins

ShNF-YA, ShNF-YB, and ShNF-YC proteins were predicted for three-dimensional structures using the Phyre2 web server “http://www.sbg.bio.ic.ac.uk/~phyre2/html/page.cgi?id=index (accessed on 15 March 2023)”.

### 2.7. RNA Extraction and Real-Time Quantitative PCR

*Saccharum* commercial hybrid Co 86032 and *E. arundinaceus* were grown in pots under glasshouse conditions with the conditions mentioned elsewhere [27,28]. Drought stress was induced in 90-day-old plants by withdrawing irrigation for 10 days. Young leaf tissues of 90-day-old plants from drought-treated *Saccharum* commercial hybrid Co 86032 and *E. arundinaceus* were collected at 5- and 10-day intervals, frozen in liquid nitrogen immediately, and stored at −80 °C. Total RNA extraction from the young leaf and root tissues was conducted using a Qiagen kit (Plant RNeasy Kit, Qiagen, Hilden, Germany). DNase I (Thermo Fisher Scientific, Lenexa, KS, USA) treatment was given to remove genomic DNA contamination from the extracted total RNA. First-strand cDNA synthesis using Revert Aid First-Strand cDNA Synthesis Kit (Thermo Fisher Scientific, Lenexa, KS, USA) was carried out in a total reaction volume of 20 µL with Dnase I-treated RNA of 1000 ng and an oligo dT _(18)_ primer.

Expression of arbitrarily selected *ShNF-Ys* and *EaNF-Ys* in leaf and root tissues under drought stress was analyzed using quantitative real-time reverse transcription (qRT)-PCR. Forward and reverse primers used for real-time quantification studies are given in Appendix A. PCR reactions were carried out with a total volume of 20 µL in the StepOne real-time PCR system (Applied Biosystems, Burlington, ON, Canada) with the following temperature profile: 10 min of denaturation at 95 °C, followed by 40 cycles of 15 s of denaturation at 95 °C, 60 min of annealing at 60 °C, and an extension of 5 min at 72 °C. The gene encoding for glyceraldehyde 3-phosphate dehydrogenase (GAPDH) was used as an internal control. Normalization of raw threshold values was performed against GAPDH. Relative expression of the NF-YB2 gene in *E. arundinaceus* and Co 86032 was determined using the 2^−∆∆Ct^ method [29]. Three technical and biological replicates were used for the expression analysis studies.

## 3. Results

### 3.1. Identification, Characterization of ShNF-Y Transcription Factors, and Conserved Domain/Motif Analysis

In sugarcane, 9 *NF-YA*, 18 *NF-YB*, and 24 *NF-YC* genes were identified using a blast search against a mosaic monoploid reference sugarcane genome. Identified *NF-Ys* were predicted for ORFs, cds, and protein sequences (Table 1). Conserved functional domains were predicted for NF-Y protein sequences using InterProScan, pfam, NCBI’s Conserved Domain Database, and MOTIF Search. All the identified ShNF-Ys had conserved core functional domains necessary for subunit interaction and DNA binding. ShNF-YA, ShNF-YB, and ShNF-YC proteins had the conserved CBF-B/HAP2/NF-YA, CBF-A/HAP3/NF-YB, and ShNF-CBF-C/HAP5/NF-YC domains, respectively(Figure 2, Figure 3 and Figure 4). *ShNF-Y* genes were predicted to have conserved motifs (Appendix A). Ten gene motifs were identified in all three *ShNF-Y* genes. ShNF-Y proteins were also predicted to have conserved motifs (Figure 5).

### 3.2. Physiochemical Properties and Subcellular Localization Analysis

ShNF-YA, ShNF-YB, and ShNF-YC proteins were predicted for physiochemical properties (molecular weight, isoelectric point, and hydropathicity) using the ExPASy Protparam tool (Table 1). ShNF-Y proteins were predicted for subcellular localization and gene ontology (Appendix A). ShNF-Y proteins were predicted to be localized in the nucleus. Most of the ShNF-Y members were annotated as CCAAT-binding factor complexes. The gene ontology annotation of the ShNF-Y members had thylakoid, vacuoles, and cytoplasm gene ontology identifiers.

### 3.3. Multiple Sequence Alignment and Phylogenetic Analysis of ShNF-Y Proteins

Multiple sequence alignment (MSA) of conserved regions of ShNF-Ys along with orthologous species of sorghum and *Arabidopsis* was carried out (Figure 2, Figure 3 and Figure 4). Alignment of NFY conserved regions depicted the conservation of amino acids involved in DNA binding and subunits interaction.

Phylogenetic analysis of NF-Ys in sugarcane, sorghum and Arabidopsis have shown that ShNF-YA subunits were equidistant from each other (Figure 6a). However, ShNF-YB and ShNF-YC subunits clustered distinctly forming closely related and divergent groups (Figure 6b,c). Phylogentic tree of all ShNF-Y proteins showed five major clades (Figure 6d).

### 3.4. Gene Structure, Chromosomal Distribution, and Synteny Analysis of ShNF-Y Genes

Gene structures for *ShNF-Y* genes were predicted. *ShNF-YA* genes had an average of four introns and the intronic regions were longer (Figure 7a). Gene lengths for ShNF-YA genes ranged from 821 to 6739 base pairs (bp). In contrast to *ShNF-YA* genes, ShNF-YB genes had more CDS regions than intronic regions. Most of the *ShNF-YB* genes lacked introns (Figure 7b). The gene length of *ShNF-YB* genes ranged from 312 to 2121 bp. *ShNF-YC* genes also had fewer intronic regions when compared to ShNF-YA genes (Figure 7c). The gene length of *ShN-YC* genes ranged from 324 to 8370 bp.

Identified *NF-Y* sequences were located on all 10 chromosomes of the sugarcane genome (Figure 8). *NF-YA* genes were distributed on chromosomes 1, 2, 4, and 8. *NF-YA1* and *NF-YA2* were located on chromosome 1. *NF-YA3* and *NF-YA4* were located on chromosome 2. *NF-YA5*, *NFYA6*, and *NF-YA7* were located on chromosome 4. *NF-YA8* and *NF-YA9* were located on chromosome 8. NF-YB genes were distributed on chromosomes 1, 2, 3, 4, 9, and 10. *NF-YC* genes were distributed on all chromosomes except for chromosome 10. *NF-YC1*, *NF-YC2*, and *NF-C3* were located on chromosome 1. *NF-YC4*, *NF-YC5*, *NF-YC6*, *NF-YC7*, *NF-YC8*, *NF-YC9*, *NF-YC10*, and *NF-YC11* were located on chromosome 2. *NF-YC12* was located on chromosome 3. *NF-YC 13*, *NF-YC 14*, and *NF-YC 15* were located on chromosome 4. *NF-YC16* was located on chromosome 5, and *NF-YC17* was located on chromosome 6. *NF-YC18*, *NF-YC19*, *NF-YC20*, *NF-YC21*, and *NF-YC22* were located on chromosome 7. *NF-YC23* was located on chromosome 8, and *NF-YC24* was located on chromosome 9.

Synteny analysis of *NF-Y* genes in sugarcane, *Arabidopsis*, and sorghum was performed to explore the evolutionary relationship of the *ShNF-Y* gene family. Sixteen orthologous gene pairs were identified in sugarcane and sorghum (Figure 9a). Orthologous gene pairs identified in sugarcane and sorghum were *ShNF-YA2/SbNF-YA4*, *ShNF-YA3/SbNF-YA10*, *ShNF-YA4/SbNF-YA5*, *ShNF-YA5/SbNF-YA5*, *ShNF-YA6/SbNF-YA5*, *ShNF-YA7/SbNF-YA5*, *ShNF-YB11/SbNF-YB2*, *ShNF-YB16/SbNF-YB3*, *ShNF-YB17/SbNF-YB3*, *ShNF-YC5/SbNF-YC2*, *ShNF-YC6/SbNF-YC3*, *ShNF-YC12/SbNF-YC4*, *ShNF-YC17/SbNF-YC5*, *ShNF-YC18/SbNF-YC6*, *ShNF-YC19/SbNF-YC6*, and *ShNF-YC22/SbNF-YC7*. Similarly, fifteen orthologous gene pairs were identified in sugarcane and *Arabidopsis* (Figure 9b). Orthologous gene pairs identified in sugarcane and *Arabidopsis* were *ShNF-YA3/AtNF-YA2*, *ShNF-YA9/AtNF-YA1*, *ShNF-YB9/AtNF-YB5*, *ShNF-YB10/AtNF-YB4*, *ShNF-YB11/AtNF-YB6*, *ShNF-YC3/AtNF-YC11*, *ShNF-YC4/AtNF-YC13*, *ShNF-YC5/AtNF-YC9*, *ShNF-YC6/AtNF-YC9*, *ShNF-YC7/AtNF-YC13*, *ShNF-YC13/AtNF-YC12*, *ShNF-YC15/AtNF-YC12*, *ShNF-YC18/AtNF-YC3*, *ShNF-YC22/AtNF-YC2 and ShNF-YC24/AtNF-YC13*. Overall, the ShNF-Ys genes consisted of more syntenic gene pairs with both monocots and dicots.

### 3.5. Three-Dimensional Structure Prediction of ShNF-Y Proteins

Three-dimensional structures were predicted for ShNF-Y protein sequences (Appendix A). ShNF-Y protein structures were dominated by helix and loop regions.

### 3.6. Expression Analysis of NF-Y Genes in the Saccharum Complex under Drought Stress

Quantification of expression of arbitrarily selected NF-Y genes (4 NF-YA, 9 NF-YB, and 5 NF-YC) revealed differential regulation in leaf and root tissues of the *Saccharum* hybrid Co 86032 and drought-tolerant *E. arundinaceus* at the 10th day of drought stress (Figure 10). NF-Y gene expression indicated a tissue-specific and drought-inducible expression profile in the *Saccharum* hybrid and *E. arundinaceus*. The qRT-PCR results correlated with those of the RNAseq DGE analysis, illustrating the reliability of the transcriptome profile data obtained. Among the 14 genes tested, a few were upregulated in both leaf and root tissues, some others exhibited downregulation, and the rest were not influenced by the stress conditions in the transcriptome data (Appendix A). A similar trend was found in the qRT-PCR analysis, with differences in the fold expression levels.

## 4. Discussion

NF-Y subunits regulate stress-responsive signaling pathways and networks by modulating downstream targets [30]. The role of NF-Y subunits in modulating epigenetic mechanisms also implies that they are crucial players in stress-defensive responses. Many reports suggest that overexpression of the NF-Y subunits confers tolerance to various stresses, including drought and salinity, in plants [13,21,30,31,32,33]. NF-Y subunits act both as complexes and as individual subunits in regulating gene expression and stress response. As complexes, NF-Y subunits modulate gene expression by binding to the CCAAT box in the promoter regions [34]. Some studies indicate NF-YB and NF-YC binding with other transcription factors in transcriptional activation mechanisms [35,36].

Owing to the importance of NF-Y transcription factors in various stress-responsive mechanisms and developmental processes, several studies have been carried out to identify *NF-Y* genes in different plant species. In soybean, 21 *NF-YA*, 32 *NF-YB*, and *15 NF-YC* genes have been identified. Certain NF-Y gene members have been identified to have a vital role in specific stress responses and developmental processes [37]. In *Triticum aestivum*, 10 *NF-YA*, 11 *NF-YB*, and 14 *NF-YC* genes have been identified. Expression analysis of *TaNF-Y* genes revealed some *NF-Y* members have ubiquitous expression, while some are organ-specific and some are drought-responsive [38]. In *S. bicolor*, 8 *NF-YA*, 11 *NF-YB*, and 14 *NF-YC* genes were identified. In silico expression analysis under salt, drought, cold, and heat stresses revealed that certain NF-Y genes are stress-responsive. In *Vitis vinifera*, 8 *NF-YA*, 12 *NF-YB*, and 8 *NF-YC* gene members were identified. Expression analysis revealed a number of *VvNF-Y* gene members involved in various biotic and abiotic stresses, phytohormone regulation, and sugar metabolism [39]. The wide expansion of NF-Y gene members in the genome highlights the essential role of these plant nuclear factors in various functions in sugarcane [40].

In this study, a genome-wide analysis of ShNF-Ys was performed, and 9 *ShNF-YA*, 18 *ShNF-YB*, and 24 *ShNF-YC* gene members were identified. In silico and real-time expression analyses of *ShNF-Ys* were performed. Physiochemical parameters, domain analysis, phylogeny, and three-dimensional structure prediction assisted in gaining insights about the ShNF-Ys.

Subcellular localization predictions of proteins helped to identify their localization in the cellular compartments, thereby helping to understand their role in the cellular machinery. This study predicted that all ShNF-Y proteins would be localized in the nucleus. For instance, the NF-YB proteins of *E. arundinaceus* (EaNF-YB2) [41], Poplar (PdNF-YB7) [42], *A. thaliana* (HAP3b) [43], and *Oryza sativa* (OsHAP3H) [44] are localized at the nucleus. The localization of foxtail millet NF-YB8 showed a cell-wide distribution pattern [24]. Similarly, *Arabidopsis* NF-YB3 was translocated to the nucleus only during stress conditions [6]. This suggests that ShNF-Y members might also localize in various cellular regions other than the nucleus and play vital roles in different cellular processes. However, experimental studies have to be carried out to ascertain the localization mechanisms of ShNF-Y proteins.

Synteny analysis showed that *NF-Y* orthologs were conserved within the syntenic blocks. Syntenic gene pairs identified between sugarcane, sorghum, and *Arabidopsis* could be exploited to understand functional equivalence between these species. Notably, 15 ShNF-Ys members were found to be syntenic with the NF-Y members across monocot and dicot species, which indicated that these orthologous pairs are conserved and may have existed before the ancestral divergence [45,46]. However, the intersections of the syntenic NF-Ys members may be valuable for exploration in evolution studies. These syntenic *NF-Y* gene pairs identified among sugarcane, sorghum, and *Arabidopsis* might also share similar expression patterns with key functional properties.

*NF-Y*s exhibit differential expression patterns under biotic and abiotic stresses, specifically drought stress in crop plants [10,47]. The differential expression pattern of *NF-Y* genes signifies their involvement in drought-responsive stress mechanisms. Tissue-specific expression of NF-Ys was recorded in some of the plant species [47,48]. Real-time expression analysis of *ShNF-Y* genes investigated the differential expression pattern in *Erianthus* and sugarcane. *NF-YA3*, *NF-YA5*, *NF-YB2*, *NF-YB8*, *NF-YB9*, *NF-YC7*, *NF-YC9*, and *NF-YC18* genes had significantly higher expression under drought, whereas all other NF-Ys were either downregulated or had non-significant expression in the *Saccharum* complex. *NF-YA*, *NF-YA1*, and *NF-YA9* were downregulated in leaf and root tissues of the *Saccharum* hybrid and *E. arundinaceus* under drought conditions. *NF-YA3* showed significantly higher expression only in the root of *E. arundinaceus*, with a fold change of 3.65. Higher expression of *NF-YA5* was recorded in leaf tissues of both *E. arundinaceus* and the *Saccharum* hybrid, as well as in the root of *E. arundinaceus*. Among the investigated *NF-YB* genes, ubiquitous higher expression of *NF-YB2* was recorded in both tissues (fold changes of 4.123, 6.123, 1.236, and 3.896 in ShL, EaL, ShR, and EaR, respectively) except in *Saccharum* hybrid root. This indicates the involvement of *NF-YB2* in the stress response. Root-specific expression was observed with the *ShNF-YB8* gene. Higher transcript levels of *ShNF-YB8* in the root suggests its plausible role in drought response. *ShNF-YB16* was universally downregulated in leaf and root tissues of both varieties. Considering *NF-YC*, *ShNF-YC7*, and *ShNF-YC9* had higher expression in leaves (a fold change of 3.8 (ShL) and 3.9 (EaL)). *NF-YC14* had downregulated expression invariably in the leaves and roots of the *Saccharum* complex. *NF-YC18* was downregulated in ShL, EaL, and ShR and upregulated (a fold change of 5.32) in EaR. Higher expression in EaR implies that *NF-YC18* is root-specific and stress-responsive in *E. arundinaceus. NF-YC24* had lower expression in *E. arundinaceus* and was downregulated in the *Saccharum* hybrid. In the *Saccharum* hybrid, *ShNF-YA5*, *ShNF-YB2*, *ShNF-YC7*, and *ShNF-YC9* were leaf-specific, and *ShNF-YB8* and *ShNF-YC18* were root-specific. In *E. arundinaceus*, elevated expression of *NF-Y* genes was observed in leaves as well as in roots. *EaNF-YC7* and *EaNF-YC9* had elevated expression in leaves. *EaNF-YA5* and *EaNF-YB2* were upregulated in both leaf and root tissues. However, *E. arundinaceus* had more root-specific expression of *NF-Y* genes compared to the *Saccharum* hybrid. *EaNF-YA5* gene expression was prominent in *E. arundinaceus* roots, suggesting its potential role in root modifications during drought stress. Besides, *EaNF-YB2* had a significantly higher fold change of expression in roots, endorsing its crucial role in stress-adaptive regulatory mechanisms. *EaNF-YA3*, *EaNF-YB8*, and *EaNF-YC18* also had higher expression in roots.

Higher transcript levels of root-specific *NF-Y* genes in *Erianthus* suggest that the root plays an essential role in drought tolerance in the wild relative. Tissue-specific expression of *NF-Y* genes highlights their role in tissue-specific responses to combat drought stress. The NF-Ys might function as individual components and/or in combination with other genes in specific tissues to impart drought tolerance in sugarcane.

Overexpression studies on *NF-Y* gene members in different plant species envisaged their role in tolerance to various types of stresses. Previous investigation on the overexpression of *AtNF-YA5* in *Arabidopsis* showed reduction in water loss and improved tolerance to drought in transgenics [13]. Likewise, transgenic rice overexpressing *NF-YA7* exhibited tolerance to drought in an ABA-independent manner [49]. *NF-YB3* from *Picea wilsonii*, when overexpressed in *Arabidopsis*, conferred tolerance to osmotic, salinity, and drought stresses [31]. Overexpression of the *NF-YC* gene in *Amaranthus hypochondriacus* conferred resistance to water-deficit stress in *Arabidopsis* [32]. *ShNF-YA3*, which had higher expression in the root, showed an identity of 88.31% with the rice nuclear transcription factor *OsHAP2E*, which upon overexpression exhibited tolerance to salinity and drought stresses. *ShNF-YA5* showed an identity of 97.29% with *TaNFYA-B1* nuclear transcription factor from *T. aestivum*, which stimulated root development upon overexpression [33]. This implies the significance of *ShNF-YA5* in root development, an essential trait for drought tolerance. Similarly, *ShNF-YC18*, which was root-specific, shared a 93.1% identity with the *A. hypochondriacus* NF-YC gene. Hence, overexpressing *ShNF-YC18* may provide drought tolerance in sugarcane. Similarly, higher expression of *EaNF-YB2* was observed in *E. arundinaceus* compared to the *Saccharum* hybrid under drought and salinity conditions [41], suggesting that overexpression of *ShNF-YB2* might assist sugarcane in developing tolerance against drought and salinity stresses. *Erianthus*, which has a well-developed root system inherently [50], showed higher expression of root-specific *EaNF-Y* genes under drought. Upregulation of *NF-Y* genes in roots shows that *NF-Y* genes have a vital role in root adaptation mechanisms for drought tolerance [31,42]. These studies suggest that overexpressing *NF-Y* genes in plants would assist in combating drought stress. Differentially expressed *NF-Y* genes under drought stress in the *Saccharum* complex can act as potential candidate genes for crop improvement.

## 5. Conclusions

Comprehensive genomic analysis and characterization of NF-Y genes in the commercial sugarcane (*Saccharum* spp.) hybrid identified 9 *NF-YA*, 18 *NF-YB*, and 24 *NF-YC* genes, and also revealed differential expression patterns of *ShNF-Y* genes in leaves and roots under drought conditions. This provides the basis for exploiting stress-responsive *NF-Y* genes as potential candidates for sugarcane genetic improvement through conventional breeding techniques and modern biotechnological approaches.

## Figures and Tables

**Figure 1 genes-14-01147-f001:**
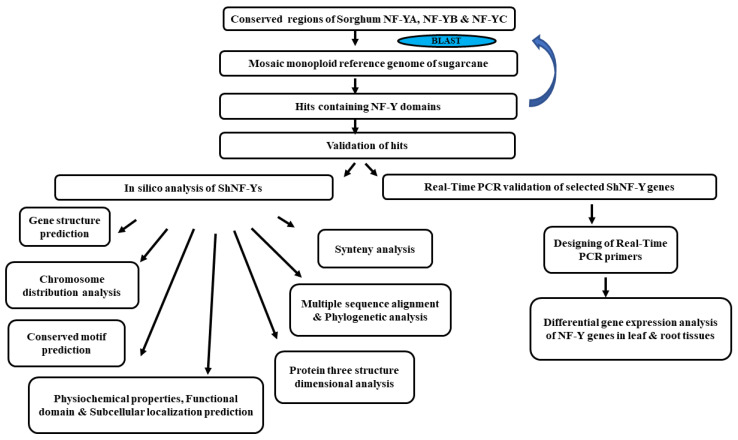
Bioinformatic pipeline for identification, in silico, and expression analysis of *NF-Y* genes.

**Figure 2 genes-14-01147-f002:**
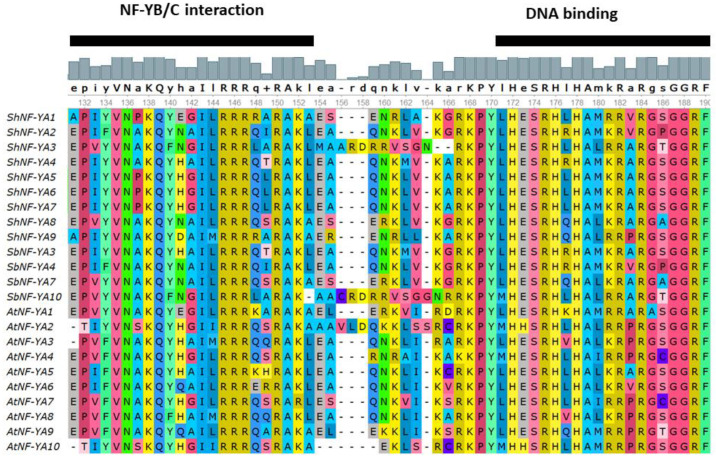
Multiple sequence alignment of NF-YA conserved domains.

**Figure 3 genes-14-01147-f003:**
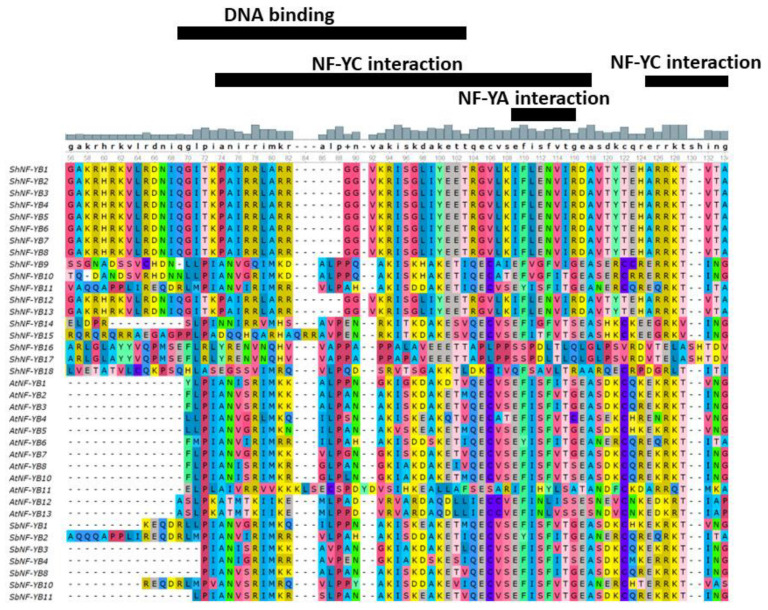
Multiple sequence alignment of NF-YB conserved domains.

**Figure 4 genes-14-01147-f004:**
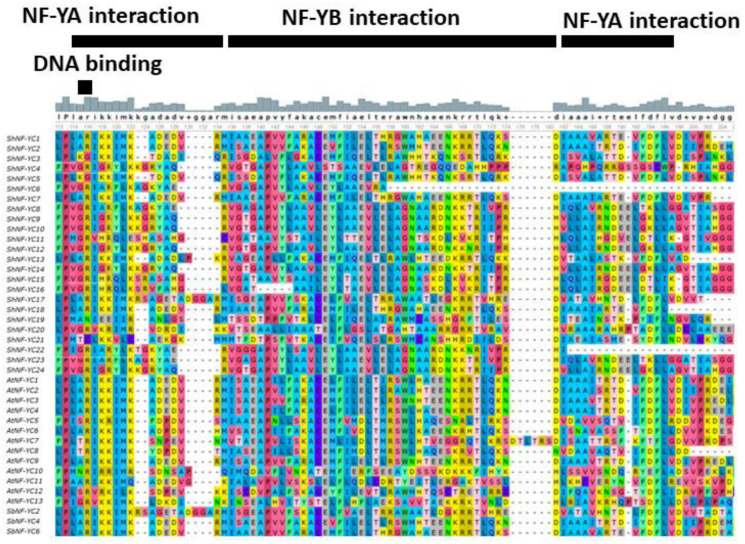
Multiple sequence alignment of NF-YC conserved domains.

**Figure 5 genes-14-01147-f005:**
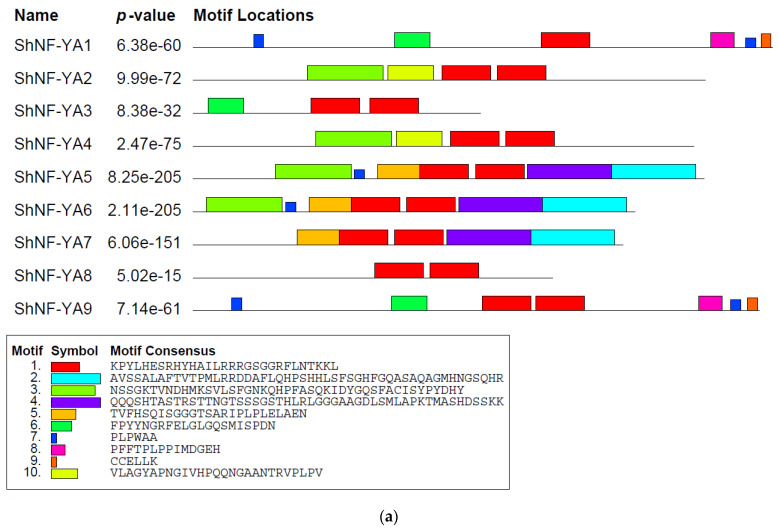
Motifs predicted in ShNF-Y proteins: (**a**) ShNF-YA; (**b**) ShNF-YB; and (**c**) ShNF-YC.

**Figure 6 genes-14-01147-f006:**
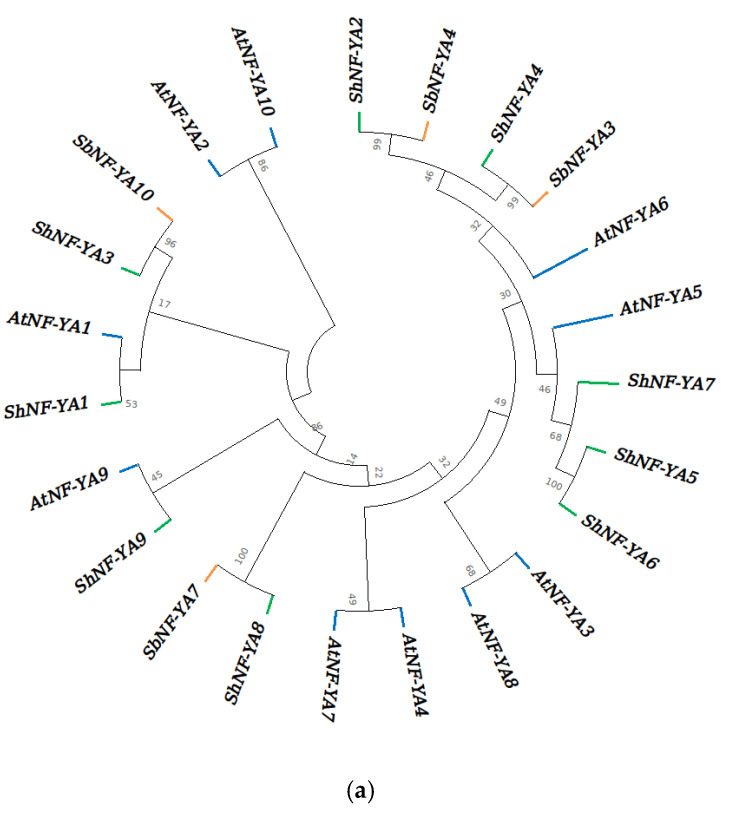
Phylogenetic tree of NF-Ys in sugarcane, sorghum, and *Arabidopsis*: (**a**) NF-YA phylogenetic tree; (**b**) NF-YB phylogenetic tree; (**c**) NF-YC phylogenetic tree; ShNF-Ys are highlighted with green solid circles; and (**d**) phylogenetic tree of all ShNF-Ys. Trees are constructed with 1000 bootstrap replicates.

**Figure 7 genes-14-01147-f007:**
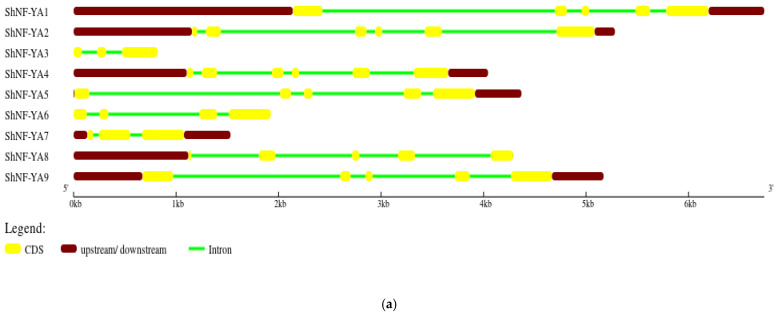
Gene structure of *ShNF-Y* genes: (**a**) *ShNF-YA*; (**b**) *ShNF-YB*; and (**c**) *ShNF-YC*.

**Figure 8 genes-14-01147-f008:**
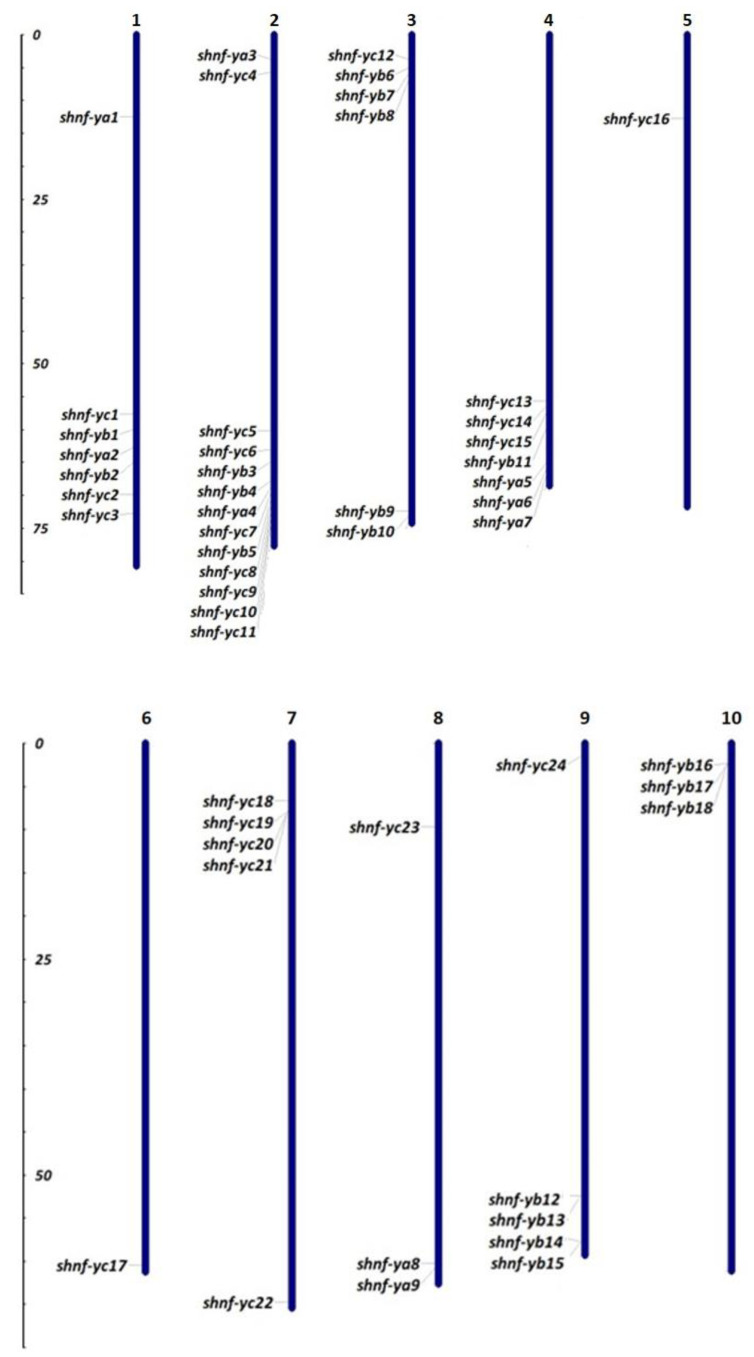
Chromosomal distribution of *NF-Y* gene members in sugarcane. The numbers at the top of the bars represent chromosome numbers. The scale on the left is in megabases (Mb).

**Figure 9 genes-14-01147-f009:**
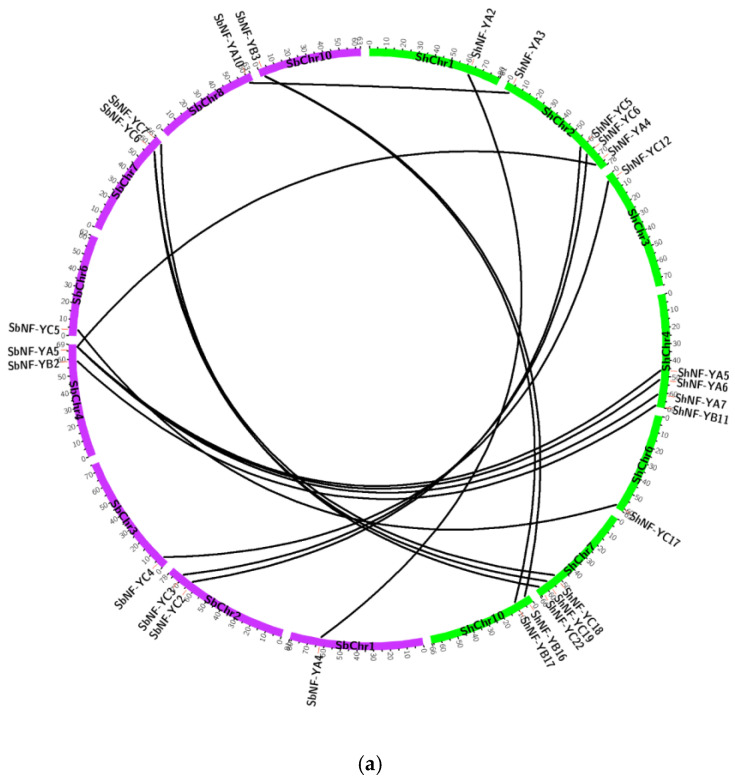
Synteny of *NF-Y* genes among sugarcane (*Saccharum* hybrid), sorghum (*Sorghum bicolor*), and *Arabidopsis thaliana*. (**a**) Synteny analysis between sugarcane and sorghum; (**b**) Synteny analysis between sugarcane and *Arabidopsis*. Chromosomes of sugarcane (ShChr), sorghum (SbChr), and *Arabidopsis* (AtChr) are colored green, purple, and blue, respectively. Lines connecting NF-Y genes represent orthologous gene pairs.

**Figure 10 genes-14-01147-f010:**
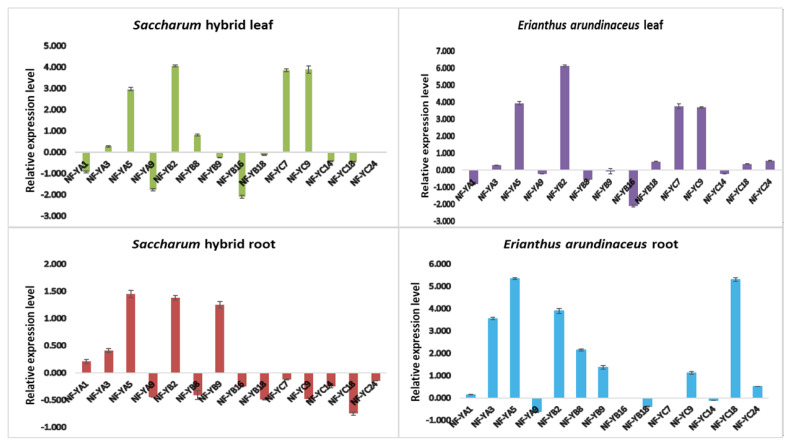
Expression of *ShNF-Y* gene members under drought in leaf and root tissues of the *Saccharum* hybrid and *Erianthus arundinaceus*.

**Table 1 genes-14-01147-t001:** Physiochemical properties of ShNF-Y subunits.

Gene Name	Cds Length	Gene Length	Protein Length	Molecular Weight (kDA)	Protein Theoretical pI	GRAVY
NF-YA Subunit						
ShNF-YA1	1038	6739	345	36.542	9.61	−0.533
ShNF-YA2	918	5282	305	33.192	9.73	−0.650
ShNF-YA3	516	821	171	19.334	11.77	−0.743
ShNF-YA4	897	4044	298	32.588	8.21	−0.848
ShNF-YA5	915	4370	304	32.894	11.18	−0.730
ShNF-YA6	792	1924	263	28.528	10.37	−0.580
ShNF-YA7	771	1531	256	28.552	10.58	−0.388
ShNF-YA8	645	4290	214	23.355	7.97	−1.101
ShNF-YA9	1014	5171	337	36.532	9.24	−0.500
NF-YB Subunit						
ShNF-YB1	312	312	103	11.409	11.48	−0.524
ShNF-YB2	312	565	103	11.409	11.48	−0.524
ShNF-YB3	312	527	103	11.409	11.48	−0.524
ShNF-YB4	312	312	103	11.409	11.48	−0.524
ShNF-YB5	312	649	103	11.409	11.48	−0.524
ShNF-YB6	312	312	103	11.409	11.48	−0.524
ShNF-YB7	312	852	103	11.409	11.48	−0.524
ShNF-YB8	312	312	103	11.409	11.48	−0.524
ShNF-YB9	501	501	166	18.324	5.95	−0.480
ShNF-YB10	501	715	166	18.363	6.52	−0.821
ShNF-YB11	783	783	260	27.691	6.40	−0.579
ShNF-YB12	312	632	103	11.409	11.48	−0.524
ShNF-YB13	312	312	103	11.409	11.48	−0.524
ShNF-YB14	510	1743	169	18.693	5.82	−0.435
ShNF-YB15	549	2121	182	20.447	9.83	−0.804
ShNF-YB16	489	489	162	17.652	4.52	−0.141
ShNF-YB17	435	435	144	15.681	4.40	−0.143
ShNF-YB18	390	390	129	13.982	9.35	0.018
NF-YC Subunit						
ShNF-YC1	555	555	184	19.855	5.50	−0.280
ShNF-YC2	765	8370	254	28.516	5.04	−0.668
ShNF-YC3	1404	2263	467	52.251	5.19	−0.262
ShNF-YC4	720	751	239	25.574	10.74	−0.478
ShNF-YC5	1173	1325	390	43.027	5.66	10.681
ShNF-YC6	417	816	138	14.617	10.76	−0.094
ShNF-YC7	567	567	188	20.032	5.36	−0.065
ShNF-YC8	507	995	168	17.849	11.63	−0.136
ShNF-YC9	480	739	159	16.482	10.68	−0.397
ShNF-YC10	471	803	156	16.294	10.68	−0.388
ShNF-YC11	324	324	107	11.491	8.67	0.110
ShNF-YC12	471	471	156	16.299	10.68	−0.312
ShNF-YC13	519	519	172	18.911	5.72	−0.641
ShNF-YC14	486	609	161	16.795	11.14	−0.346
ShNF-YC15	423	1263	140	14.813	10.25	−0.386
ShNF-YC16	594	2681	197	22.139	10.34	−0.120
ShNF-YC17	384	384	127	13.454	6.05	−0.207
ShNF-YC18	609	973	202	21.449	5.37	−0.130
ShNF-YC19	462	462	153	17.293	5.13	−0.448
ShNF-YC20	423	423	140	14.765	10.54	−0.246
ShNF-YC21	606	600	201	23.414	8.14	−0.636
ShNF-YC22	255	1312	84	8.882	10.62	−0.360
ShNF-YC23	405	893	134	13.957	10.05	−0.181
ShNF-YC24	474	739	157	16.288	10.68	−0.314

## Data Availability

All data generated or analyzed during this study have been included in this article.

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
