# Peer review of "Genome-Wide Identification, Characterization and Expression Analysis of Plant Nuclear Factor (NF-Y) Gene Family Transcription Factors in Saccharum spp."

_genes, 2023, doi:10.3390/genes14061147_

Round 1

Reviewer 1 Report

This study reports on a systematic study of the plant nuclear factor (NF-Y) gene subfamily in sugarcane. The authors identified 51 NF-Y genes in sugarcane and performed multiple sequence alignment of ShNF-Y proteins, identified orthologous gene pairs between sugarcane and sorghum, and conducted phylogenetic analysis of NF-Y subunits of sugarcane, sorghum, and Arabidopsis. The authors also investigated the expression profiling of NF-Y gene members under drought treatment and found that some NF-Y genes were involved in drought tolerance in Saccharum hybrid and its drought tolerant wild relative Erianthus arundinaceus. The results of this study provide valuable genetic resources for further sugarcane crop improvement programs. The authors' systematic approach to identifying and characterizing NF-Y genes in sugarcane contributes to our understanding of the functions of NF-Y transcriptional factors in plants, particularly under different developmental and stress conditions. However, further studies are needed to explore the regulatory mechanisms of NF-Y genes in sugarcane and their potential application in crop improvement programs.

Here are my comments and suggestions:

1.  A workflow of the whole study including expression analysis based on PCR validation is suggested to be shown in Figure 1, instead of only bioinformatic pipeline and in silico analysis. 

2. In addition to Figure 7, all ShNF-YA, B, and C genes are suggested to be used to conduct a phylogenetic tree.

3. The authors should revise the language and visualization of this manuscript to improve readability. Those papers could be a reference (such as https://doi.org/10.3389/fpls.2019.00565 and https://doi.org/10.3390/genes10120980)

4. In figure 11, why are those few ShNF-Y proteins selected? Three dimensional structures models of the other proteins should be provided in supplementary materials.

The authors should revise the language and visualization of this manuscript to improve readability. 

Author Response

Response Sheet – Reviewer 1

Comments and Suggestions for Authors

This study reports on a systematic study of the plant nuclear factor (NF-Y) gene subfamily in sugarcane. The authors identified 51 NF-Y genes in sugarcane and performed multiple sequence alignment of ShNF-Y proteins, identified orthologous gene pairs between sugarcane and sorghum, and conducted phylogenetic analysis of NF-Y subunits of sugarcane, sorghum, and Arabidopsis. The authors also investigated the expression profiling of NF-Y gene members under drought treatment and found that some NF-Y genes were involved in drought tolerance in Saccharum hybrid and its drought tolerant wild relative Erianthus arundinaceus. The results of this study provide valuable genetic resources for further sugarcane crop improvement programs. The authors' systematic approach to identifying and characterizing NF-Y genes in sugarcane contributes to our understanding of the functions of NF-Y transcriptional factors in plants, particularly under different developmental and stress conditions. However, further studies are needed to explore the regulatory mechanisms of NF-Y genes in sugarcane and their potential application in crop improvement programs.

Here are my comments and suggestions:

  1. A workflow of the whole study including expression analysis based on PCR validation is suggested to be shown in Figure 1, instead of only bioinformatic pipeline and in silico analysis. 

Response: Agreed. Workflow is revised and included the expression analysis in addition to bioinformatic pipeline and in silico analysis

  1. In addition to Figure 7, all ShNF-YA, B, and C genes are suggested to be used to conduct a phylogenetic tree.

Response: All ShNF-YA, B and C genes were included and prepared the single phylogenetic tree and incorporated in the manuscript main text. However, the individual phylogenetic tree of ShNF-YA, B, and C genes are given as supplementary figures.

  1. The authors should revise the language and visualization of this manuscript to improve readability. Those papers could be a reference (such as https://doi.org/10.3389/fpls.2019.00565 and https://doi.org/10.3390/genes10120980)

Response: Thank the reviewer, English language is revised and indicated in the manuscript with track change option. By following the reference papers suggested by the reviewer, figures are prepared afresh and included in the manuscript to improve the visualization.

  1. In figure 11, why are those few ShNF-Y proteins selected? Three dimensional structures models of the other proteins should be provided in supplementary materials.

Response: Yes, dimensional structures models of the other proteins were given in the supplementary files. Now, we wish to include it in the main text itself in order to visualise the difference among the proteins of ShNF-Ys.

Comments on the Quality of English Language. The authors should revise the language and visualization of this manuscript to improve readability. 

Response: Thank the reviewer, English language is revised and indicated in the manuscript with track change option. By following the reference papers suggested by the reviewer, figures are prepared afresh and included in the manuscript to improve the visualization.

Reviewer 2 Report

I found the manuscript difficult to read especially because it lists a series of in silico experiments that need better discussion and justification. Requires polishing and presenting the most relevant results to the reader. Please focus on the parts that you think the reader should focus and why these are important. For example: Table 1 contains GRAVY scores but why should care about these? Why some of the proteins have positive scores - do they have different properties than the other proteins? Scientific writing is about making further connections and providing insight.

I found the Intro and the Discussion minimal while the results section is too overwhelming. Try to enrich especially the Discussion which does not address your major findings. 

Several method sections require improvement. For example - Figure 11 Y1A contains unstructured protein segments which is rather a poor prediction and I don't think it needs to be added to the main text. Have you tried to use AlphaFold/RosettaFold tor these? Can you please add quality scores to your predicted structures?

Analysis in 3.3 is a list of results that is difficult to read. I recommend adding more to the synteny analysis rather than just pointing to the obvious comparisons. For example: can you find other genes that are neighboring NF-Y? What is the importance of the synteny results you are presenting?

Figures 2, 3 4: please avoid this type of figures in the main text. Add these to the Supplementary and possibly present ONE figure with the info you want to convey which is the domains annotation

Figures 5, 6: Same as comment before. Can you polish and consolidete their message?

Table 1: Protein length and CDS length are correlated values. length aa * 3 = lenght cds. Could you please check if your definition of CDS and amminoacid length is correct? SNF-YA1 has a longer amminoacid sequence than SNF-YA2 but this is not reflected in the CDS.

I appreciated the qPCR validation which is a suitable experiment for this type of manuscript. Do you also have any RNAseq data (also from SRA public datasets) that you can include as well?

The English language is suitable for scientific writing

Author Response

Response Sheet – Reviewer 2

Comments and Suggestions for Authors

I found the manuscript difficult to read especially because it lists a series of in silico experiments that need better discussion and justification. Requires polishing and presenting the most relevant results to the reader. Please focus on the parts that you think the reader should focus and why these are important. For example: Table 1 contains GRAVY scores but why should care about these? Why some of the proteins have positive scores - do they have different properties than the other proteins? Scientific writing is about making further connections and providing insight.

Response: Suggestions are agreed. Improved the content in this revised manuscript and same is indicated with track change option.

I found the Intro and the Discussion minimal while the results section is too overwhelming. Try to enrich especially the Discussion which does not address your major findings. 

Response: Agreed. Apart from bioinformatics analysis, major finding of this study is tissue specific stress responsive expression of NF-Ys. Now the discussion part is moved from results to discussion section. Thank the reviewer for the critical constructive comments.

Several method sections require improvement. For example - Figure 11 Y1A contains unstructured protein segments which is rather a poor prediction and I don't think it needs to be added to the main text. Have you tried to use AlphaFold/RosettaFold tor these? Can you please add quality scores to your predicted structures?

Response: Agreed. As per the suggestions, this structural prediction is moved to supplementary section from main text.

Analysis in 3.3 is a list of results that is difficult to read. I recommend adding more to the synteny analysis rather than just pointing to the obvious comparisons. For example: can you find other genes that are neighboring NF-Y? What is the importance of the synteny results you are presenting?

Response: Agreed. Synteny analysis showed that NF-Ys regions of Saccharum are conserved and different from other plants.

Figures 2, 3 4: please avoid this type of figures in the main text. Add these to the Supplementary and possibly present ONE figure with the info you want to convey which is the domains annotation

Response: We wish to keep these improved version of figures in the main text as it gives information on close association and interaction among them.  

Figures 5, 6: Same as comment before. Can you polish and consolidete their message?

Response: Agreed. Motif prediction using protein is kept in the main manuscript (Figure 6) and motif prediction based on gene is given in supplementary file.

Table 1: Protein length and CDS length are correlated values. length aa * 3 = lenght cds. Could you please check if your definition of CDS and amminoacid length is correct? SNF-YA1 has a longer amminoacid sequence than SNF-YA2 but this is not reflected in the CDS.

Response: Agreed. Revised table is included in the revised manuscript.

I appreciated the qPCR validation which is a suitable experiment for this type of manuscript. Do you also have any RNAseq data (also from SRA public datasets) that you can include as well?

Response: Agreed. RNAseq data set is given in supplementary file.

Comments on the Quality of English Language

The English language is suitable for scientific writing

Response: Thank you for the point. Still, we have improved the English language by considering comments from other reviewer.

Round 2

Reviewer 2 Report

Thank you for adding the corrections and improving the manuscript. Please address the remaining suggestions:

Several method sections require improvement. For example - Figure 11 Y1A contains unstructured protein segments which is rather a poor prediction and I don't think it needs to be added to the main text. Have you tried to use AlphaFold/RosettaFold for these? Can you please add quality scores to your predicted structures?

1) Please provide info on the highlighted

I appreciated the qPCR validation which is a suitable experiment for this type of manuscript. Do you also have any RNAseq data (also from SRA public datasets) that you can include as well? 

2) I don't understand what RNAseq data set you used for the validation, could you please add the ID in GEO or SRA that corresponds to that? Please include it in the methods

3) Could you please see what happened in one of your annotations in the MSA Figure 4?  YC6 has a very long insertion which is probably due to misannotation. Can you please check it manually?

Additional comments:

* Please improve the quality of your images, and make a better resolution (Former figure 9 is not readable). For the former picture 12 - please improve the quality, and avoid overlapping labels. The first and last bar of the plot is cut, please resize it properly to fit in the area of the plot. 

!!! IMPORTANT - Please, please double-check the results that you are presenting in the main text, go over all the tables and results, and double-check manually. Be sure you are presenting relevant results and not pipeline errors. 

The English language is suitable, the paper is well-written

Author Response

Response Sheet – Reviewer 2

Comments and Suggestions for Authors

I found the manuscript difficult to read especially because it lists a series of in silico experiments that need better discussion and justification. Requires polishing and presenting the most relevant results to the reader. Please focus on the parts that you think the reader should focus and why these are important. For example: Table 1 contains GRAVY scores but why should care about these? Why some of the proteins have positive scores - do they have different properties than the other proteins? Scientific writing is about making further connections and providing insight.

Response: Suggestions are agreed. Improved the content in this revised manuscript and same is indicated with track change option.

I found the Intro and the Discussion minimal while the results section is too overwhelming. Try to enrich especially the Discussion which does not address your major findings. 

Response: Agreed. Apart from bioinformatics analysis, major finding of this study is tissue specific stress responsive expression of NF-Ys. Now the discussion part is moved from results to discussion section. Thank the reviewer for the critical constructive comments.

Several method sections require improvement. For example - Figure 11 Y1A contains unstructured protein segments which is rather a poor prediction and I don't think it needs to be added to the main text. Have you tried to use AlphaFold/RosettaFold tor these? Can you please add quality scores to your predicted structures?

Response: Agreed. We have redone the analysis and improved the prediction. As per the suggestions, this structural prediction is moved to supplementary section from main text.

Analysis in 3.3 is a list of results that is difficult to read. I recommend adding more to the synteny analysis rather than just pointing to the obvious comparisons. For example: can you find other genes that are neighboring NF-Y? What is the importance of the synteny results you are presenting?

Response: Agreed. Synteny analysis showed that NF-Ys regions of Saccharum are conserved and different from other plants.

Figures 2, 3 4: please avoid this type of figures in the main text. Add these to the Supplementary and possibly present ONE figure with the info you want to convey which is the domains annotation

Response: We wish to keep these improved version of figures in the main text as it gives information on close association and interaction among them.  

Figures 5, 6: Same as comment before. Can you polish and consolidete their message?

Response: Agreed. Motif prediction using protein is kept in the main manuscript (Figure 6) and motif prediction based on gene is given in supplementary file.

Table 1: Protein length and CDS length are correlated values. length aa * 3 = lenght cds. Could you please check if your definition of CDS and amminoacid length is correct? SNF-YA1 has a longer amminoacid sequence than SNF-YA2 but this is not reflected in the CDS.

Response: Agreed. Revised table is included in the revised manuscript.

I appreciated the qPCR validation which is a suitable experiment for this type of manuscript. Do you also have any RNAseq data (also from SRA public datasets) that you can include as well?

Response: Agreed. RNAseq data is not published yet. Hence, RNAseq data set for selected genes only given in supplementary file.

Comments on the Quality of English Language

The English language is suitable for scientific writing

Response: Thank you for the point. Still, we have improved the English language by considering comments from other reviewer.
